# Habitat Elevation Shapes Microbial Community Composition and Alter the Metabolic Functions in Wild Sable (*Martes zibellina*) Guts

**DOI:** 10.3390/ani11030865

**Published:** 2021-03-18

**Authors:** Lantian Su, Xinxin Liu, Guangyao Jin, Yue Ma, Haoxin Tan, Muhammed Khalid, Martin Romantschuk, Shan Yin, Nan Hui

**Affiliations:** 1School of Agriculture and Biology, Shanghai Jiao Tong University, Dongchuan Road 800, Shanghai 200240, China; lanter_shawn666@sjtu.edu.cn (L.S.); hostage@sjtu.edu.cn (H.T.); kokub32@gmail.com (M.K.); yinshan@sjtu.edu.cn (S.Y.); 2Instrumental Analysis Center, Shanghai Jiao Tong University, Dongchuan Road 800, Shanghai 200240, China; xinxin.liu@sjtu.edu.cn; 3Institute of Wild Animals, Heilongjiang Academy of Forestry, Haping Road 134, Harbin 150081, China; changzhao@sjtu.edu.cn; 4College of Wildlife and Protected Area, Northeast Forestry University, Hexing Road 26, Harbin 150040, China; mayue@nefu.edu.cn; 5Faculty of Biological and Environmental Science, University of Helsinki, Niemenkatu 73, 15140 Lahti, Finland; Martin.romantschuk@helsinki.fi; 6Shanghai Yangtze River Delta Eco-Environmental Change and Management Observation and Research Station, Ministry of Science and Technology, Ministry of Education, 800 Dongchuan Rd, Shanghai 200240, China; 7Shanghai Urban Forest Ecosystem Research Station, National Forestry and Grassland Administration, 800 Dongchuan Rd., Shanghai 200240, China

**Keywords:** *Martes zibellina*, gut microbial community, 16S rRNA gene, habitat environment, altitude changes

## Abstract

**Simple Summary:**

Environmental changes of habitat shaped the sable (Carnivora Mustelidae *Martes zibellina*) gut microbial community structure and altered the functions of gut microbiota, showing that the wild sable gut microbial community diversity was resilient and responded to environment change. Elevated habitat is a pivotal factor for wild sable survival and reproduction, and the adaptability is in part enabled through their gut microbial communities. Our observations show that despite having been forced to migrate from low altitudes to high altitudes because of anthropogenic habitat encroachment, wild sables showed robustness in adapting to harsh conditions. Additionally, we propose that the crucial factor enabling wild sables to survive in changeable environments was their gut microbial communities. It is widely understood that harsh conditions, such as high altitude and low temperature environments, have an adverse effect on wild fauna survival. However, our results suggested that increasing altitude can enhance some functions in wild sable gut microbial communities.

**Abstract:**

In recent decades, wild sable (Carnivora Mustelidae *Martes zibellina*) habitats, which are often natural forests, have been squeezed by anthropogenic disturbances such as clear-cutting, tilling and grazing. Sables tend to live in sloped areas with relatively harsh conditions. Here, we determine effects of environmental factors on wild sable gut microbial communities between high and low altitude habitats using Illumina Miseq sequencing of bacterial 16S rRNA genes. Our results showed that despite wild sable gut microbial community diversity being resilient to many environmental factors, community composition was sensitive to altitude. Wild sable gut microbial communities were dominated by Firmicutes (relative abundance 38.23%), followed by Actinobacteria (30.29%), and Proteobacteria (28.15%). Altitude was negatively correlated with the abundance of Firmicutes, suggesting sable likely consume more vegetarian food in lower habitats where plant diversity, temperature and vegetation coverage were greater. In addition, our functional genes prediction and qPCR results demonstrated that energy/fat processing microorganisms and functional genes are enriched with increasing altitude, which likely enhanced metabolic functions and supported wild sables to survive in elevated habitats. Overall, our results improve the knowledge of the ecological impact of habitat change, providing insights into wild animal protection at the mountain area with hash climate conditions.

## 1. Introduction

As a predator with a broad dietary diversity, the sable (Carnivora Mustelidae *Martes zibellina*) plays a crucial role in controlling food web and biodiversity in a sub-frigid forest ecosystem [1]. Sables are sensitive to habitat changes, they often serve as early warning signals regarding forest ecosystem health, and an indicator for forest management [2]. To obtain precious fur, wild sables have been overhunted in natural forest in the middle of the previous century [3]. In addition, habitats of wild sables, e.g., forest ecosystem, have been severely damaged by human activities in recent decades [4]. Part of the flat forest land was clear-cut and converted for tilling and grazing, which largely disturbed and encroached sable habitats [5]. Since the coniferous forests have been intensively affected by anthropogenic activities, the forest ecosystems were often fragmented. As forested low-altitude areas have shrunken, the distribution range of available habitats move towards higher altitudes. Thus, wild sables have been forced to migrate to higher-altitude areas and adapt to new environments.

Currently, the sable is classified as a wildlife under first class protection in China [6,7]. Several studies proposed that environmental protection and restoration of sable habitats are essential in maintenance of sable community population [8]. It is necessary to determine how environmental conditions affect sables. Previous studies demonstrated that sable population in different climatic habitats differed in various characteristics [9], and much effort has been paid to sable morphology [4,10,11], behavior [12,13,14,15], geographic distribution [16,17,18], and habitat ecology [19,20]. Characteristics of intestinal microflora, such as energy metabolism, are essential for adapting to harsh environmental conditions [21]. Yet the gut microbial communities of wild sables remain poorly characterized, especially the relationship between their intestinal microbiome function and their living environment under natural conditions remain largely unknown.

The gut microbiota, a large, complex and diverse microbial community contained in animal gastrointestinal tracts, is widely recognized as a partner that forms coevolutionary association with its host [22,23,24]. Moreover, the microbiome in animal intestines also facilitates host’s nutritional acquisition, immune modulation and homeostasis in response to profound lifestyle changes [25,26,27]. Some studies illustrate that interactions between physiological characteristics and the intestinal metagenome function of an individual are attributed to bacterial community diversity or some specific bacterial groups, e.g., species or functional clusters [28,29]. Previous studies have suggested associations between intestinal metagenome and environmental fluctuation, for example, the gut microbial community changes with habitat fragmentation and degradation [30,31,32], altitude [33,34], season [35], and climate change [36,37] of locations where the hosts live. Cold acclimated laboratory mice experience a dramatic change in gut microbiota composition as temperature increases [21]. The wild passerine gut microbial communities vary greatly with the gradient of urbanization [38], and the bacterial composition in polar bear (*Ursus maritimus*) gastrointestinal tract will change in relation to an onshore and off-shore habitat [37]. Changes in altitude, temperature, and vegetation type of host habitat may cause changes in food sources and differences in gut microbes, which in turn affect metabolic processes and ability of animals to adapt the environment. Therefore, screening microbial communities in wild sable intestine may shed some light in determining how sable can survive in harsh climate areas with cold temperature conditions.

Despite the interactions between gut microbiota and environmental factors being crucial for host survival and health, little is known regarding the environment conditions that affect the intestinal microbiome of wild sables. Previously, Zhang [39] has explored the composition of intestinal parasites and bacteria of two rescued sables using Illumina MiSeq platform. The comparison of intestinal microbiome between wild and captive sables has also been performed before [40]. It is noteworthy that almost nothing is known on how environment factors affect the sable gut microbiome. In this study, we focus on interaction between intestinal microbiome of sables and their habitat environmental utilizing 16S rRNA gene sequencing of wild sable fecal samples. We hypothesize that altitude shapes sable gut microbial community, because temperature, vegetation coverage, and nutritional sources are often associated with altitude in mountain forests; Furthermore, we also hypothesize that functional enhancements may occur in the gut microbes of wild sables in harsher condition such as higher altitude area, because gut microbial communities tend to assist the hosts to adapt to the environment changes so that they can survive.

## 2. Materials and Methods

### 2.1. Study Sites

Our study area, a typical subarctic zone, is located in the Xuexiang National Park (44° 34’ N, 128° 21’ E) in north Jilin, China (Figure 1). The area typifies temperate monsoon climate with an annual mean temperature of 3.7 °C (http://data.cma.cn/data/weatherBk.html, accessed on 12 October 2018). During sampling period December 5–25, 2018, the field temperature was around −5 to −15 °C at the foot of the mountain and dropped to a minimum of −35 °C at the top. The elevation range of this area is between 921 m to 1419 m (from river level to the highest position) also with slopes of 15° to 25°. Average altitude around each patch was calculated within a 500 m radius of the sampling point using Google Earth (earth.google.com, accessed on 23 October 2018). The main vegetation type is coniferous and broad-leaved mixed forest in Xuexiang National Park. The forests are dominated by trees of Fenghua (*Betula costata*), spruce (*Picea asperata*), diversifolious poplar (*Populus euphratica*) and fir (*Abies fabri*), shrubs of honeysuckle (*Lonicera japonica*), privet (*Ligustrum obtusifolium*) and forsythia (*Forsythia suspensa*), and herbs of Ginseng (*Panax ginseng*), cocklebur (*Xanthium sibiricum*) and soybean (*Glycine soja*). Common animals mainly include cold adaptive species, for example carnivores (sable (*Martes zibellina*), boar (*Sus scrofa*) and gluton (*Gulo gulo*)) and herbivores (roe deer (*Capreolus pygargus*), hare (*Lepus timidus*), moose (*Alces alces*), grouse (*Tetrao parvirostris*) and hazel grouse (*Bonasa bonasia*)). In this area, as the altitude increases, the vegetation coverage and canopy density gradually decrease, and the vegetation type transitions from broad-leaved forest to coniferous forest. Terrain is higher in the south and lower in the north, thus the Gudong river flows from south to north. The relatively high altitude and harsh climate result in extreme conditions for animals. Timber harvesting in Xuexiang was active from the 1950 s to the 2000 s. Part of the flat forest land was clear-cut and converted for tilling and grazing, which largely disturbed and encroached sable habitats. These human activities resulted in scattered forests mainly located in slope areas, where lands are not suitable for agriculture use but it become the new habitat for wild sables. Therefore, wild animals, such as sables, were forced to migrate to a higher altitude area and adapt to a new habitat.

### 2.2. Fecal Sample Collection

Wild sable gut was classified by sable footprint (on snow), as well as color and shape. Frozen fecal samples (*n* = 10) were collected using sterilized 50 mL falcon tubes and 70% alcohol-sterilized forceps in Xuexiang National Park along the Gudong River from south to north (Table 1). The ten samples were collected in 10 discrete forest patches with at least 500 m between them. The average altitude, vegetation coverage, canopy density and hiding cover of the forest patches were determined at a range of 500 m from the sampling spot. Average altitudes of our sampling sites ranged from 925 m to 1384 m, which covered over 80% of vertical distance in the area. The fecal samples that collected from about 950 m were classified as low-altitude group, and those from about 1350 m were classified as high-altitude group. The fecal samples were transported to laboratory in an incubator with ice pack within 24 h of collection and stored at −20 °C until DNA extraction. We brought no toxic substance that would interfere with the animal habitats. The research complied with the protocols established by the China Wildlife Conservation Association and the legal requirements of China.

### 2.3. DNA Extraction

Total DNA was extracted from approximately 0.5 g of fecal using the Fast DNA SPIN extraction kits (MP Biomedicals, Santa Ana, CA, USA), following the manufacturer’s instructions. The DNA yield was visualized by agarose gel electrophoresis (1.0% 1 × TAE buffer agarose gel run at 120 V for 1 h) with ethidium bromide. The quantity and quality of extracted DNA were measured using a NanoDrop ND-1000 spectrophotometer (Thermo Fisher Scientific, Waltham, MA, USA). The extracted DNA was stored at −20 °C before PCR amplification.

### 2.4. 16S rRNA Gene Amplification and Sequencing

PCR amplification of the V3-V4 region of the bacterial 16S rRNA genes was performed using forward primer 338F 5′-ACTCCTACGGGAGGCAGCA-3′ and the reverse primer 806R 5′-GGACTACHVGGGTWTCTAAT- 3′ [33,41]. Sample-specific 7 bp barcodes were incorporated into the primers for multiplex sequencing. We amplified the bacterial 16S rRNA genes using a two-step PCR protocol following “16S rRNA Sequencing Library Preparation” (Protocol 15044223B, Illumina). PCR amplicons were purified with Agencourt AMPure Beads (Beckman Coulter, Indianapolis, IN, USA) and quantified using the PicoGreen dsDNA Assay Kit (Invitrogen, Carlsbad, CA, USA). After the individual quantification step, amplicons were pooled in equal amounts, and paired-end 2 × 300 bp sequencing was performed using the Illlumina MiSeq platform with MiSeq Reagent Kit v3 at Shanghai Personal Biotechnology Co., Ltd. (Shanghai, China). Negative controls were included throughout the PCR and sequencing steps. The paired fastq files are available in the Sequence Read Archive at the National Center for Biotechnology Information (www.ncbi.nlm.nih.gov, accessed on 22 December 2019) under accession numbers SRX8540361-SRX8540370.

### 2.5. DNA-Based qPCR Assay

The total bacterial 16S rRNA genes were quantified using LightCycler 96 qPCR machine (roche Life Science, Beijing, China) and DyNAmo SYBR Green QPCR kit (Thermo Scientific ™, Waltham, MA, USA). The amplifications were performed by the 338F and 806R primers, used also for MiSeq amplicon generation, with 2.0 μL of diluted DNA (dilution of 1:100), 10 μL of 2X DYANAMO Master Mix, 1 μL of each primer (10 μM), and 6 μL of sterile distilled water. The thermal cycling conditions followed [42]. A negative control (dH2O) and a qPCR assay standard of Cupriavidus necator pJP4 (DSM 4058, complete genome length: 7,255,290 bp) were included. The target gene copies per genome was analyzed via the ribosomal RNA operon copy number database (rrn DB) [43]. The target gene copy numbers of the stock DNA solution were checked with the online DNA Copy Number and Dilution Calculator (Thermo Scientific™) with the known complete genome length, measured stock DNA concentration and the target gene copies per genome.

We analyzed *lpxB*, *mlaE,* and *uidA* genes using DNA cloning method. We amplified these genes from gut samples with the primers listed in Appendix A. The PCR products were purified following the manufacturers’ instructions, cloned and transformed into competent E. coli cells according to the instructions of the Zero Blunt TOPO PCR Cloning Kit for Sequencing. For quality control, all standards were linear, with R^2^ > 0.935, Error < 0.25 and Efficiency 95–105%. The Cq Error between replicates of a sample was always less than 0.5. The cloned genes were Sanger sequenced (3730, Applied Biosystems™, Foster, CA, USA) and compared to the NCBI BLAST database. The plasmids were purified following the instructions in GeneJET Plasmid Miniprep Kit (Thermo Scientific ™, Waltham, MA, USA), ran through agarose gel electrophoresis and extracted from gel using the instructions on GeneJET Gel Extraction Kit (Thermo Scientific ™, Waltham, MA, USA). Quantitative PCR assays were carried out using a LightCycler 96 qPCR machine (roche Life Science, Beijing, China) using qPCR programs and reagents listed in Appendix A.

### 2.6. Bioinformatics

Paired-end sequence data (.fastq) were processed using mothur version 1.43.0 [44]. Any sequences with ambiguous bases, with more than one mismatch to the primers, with homopolymers longer than 8 bp [45], and any without a minimum overlap of 50 bp were removed. The fasta files were aligned against the Greengene reference, pre-clustered to remove erroneous reads [46] and screened for chimeras with the Vsearch algorithm [47], and non-chimeric sequences were assigned to taxa using the naive Bayesian classifier [48] against the Greengene 13.5 training set. Non-target sequences (mitochondria, chloroplast, archaea) were removed. The final data consisted of 474,052 sequences (47,405 ± 4872 per sample, mean ± se) prior to calculating a pairwise distance matrix. Sequences were clustered to OTUs at 97% similarity using nearest neighbor (single linkage) joining. Global singelton OTUs (≤ 1 sequences across all 10 samples; 2.93% of total reads) were removed, as they may be PCR or sequencing artefacts [49]. Biological observation matrix (BIOM) was generated by referring Greengene map 13.5 [50], then 3 levels of functional gene benchmark based on KEGG Ontology were predicted using PICRUSt (version 1.1.4, Cambridge, MA, USA) [51].

### 2.7. Statistical Analysis

All statistical analyses were performed in Python (version 3.7.7, Dover, DE, USA) and R (version 3.6.3, R Core Team, Vienna, Austria) using various packages. To obtain the bacterial composition and function profiles of sable intestinal microbiome, the proportion of microbes at different taxonomic levels and functional gene benchmark were calculated in the NumPy and Pandas packages in Python. The R basic function plot and barplot were carried out for the visualization of bacterial composition and functional group distribution. The OTU (Operational Taxonomic Unit) richness, Chao1 index (1), Simpson index (2), Shannon–Weiner index (3), and Pielou evenness (4) were obtained using vegan package in R [52], and the alpha indices were also involved follow-up correlation analyses. In order to determine the relationship within sable gut microbial communities and metabolic functions, dissimilarity indices that have a rank order relation to ordering sites along gradients based on Bray-Curtis dissimilarity matrix were calculated using function *metaMDS* in vegan package. The environmental factors, including longitude, latitude, altitude, snow depth, vegetation coverage, canopy density, hiding cover and fallen trees, were correlated with the community structure and functional groups as the vector fitting procedure using permutation test by *envfit* function (permutations = 99,999) in the same analysis. Eventually, non-metric multidimensional scale (NMDS) was performed for bacterial and functional data sets to visualize the results. The correlation coefficient between alpha diversity indices, relative abundance of top 40 genera, proportion of top 40 functional genes and environmental factors were obtained using R basic function cor.test. Those with significant (*p* < 0.05) correlation were selected to visualized with heatmaps using pheatmap package, so that the relationship between them could discover more intuitively. To further confirm the influence by altitude, *t*-test between these two groups (high and low altitudes) for the means of major phyla, genera, KEGG (Kyoto Encyclopedia of Genes and Genomes) pathway modules and functional genes were also performed using R basic function *t*-test. In addition to correlation analyses, the linear regression of altitude and genera and functional genes was also performed in R basic function *lm* and visualized with scatter plot using ggplot2 package [53], due to altitude was a crucial environmental factor that had a major impact on gut microbiota.
(1)$$S_{chao1}\;={\;S}_{obs}+\frac{n_{1}\left(n_{1}-\;1\right)}{2\left(n_{2}\;+\;1\right)}$$
where S*_chao_*_1_ is Chao1 index, the predicted population number of the microbial communities based on observation number, S*_obs_* is the observation species number, *n*_1_ is the number of OTUs that contain only 1 read, *n*_2_ is the number of OTUs that contain 2 reads.
(2)$$D\;=\;1-\sum P_{i}^{2}$$
where D is Simpson index, one of the biodiversity indices in microbial ecology (the greater the value of D, the higher the biodiversity), *P_i_* is the proportion of species *i* in the total community.
(3)$$H\;=\;-\sum P_{i}\;ln\;P_{i}$$
where H is Shannon-Weiner index, another biodiversity indices in microbial ecology (the greater the value of H, the higher the biodiversity), *P_i_* is the proportion of species *i* in the total community.
(4)$$E\;=\;\frac{H}{ln{\;S}_{obs}}$$
where E is Pielou evenness, an index that reflects the uniformity of microbial species distribution (the greater the value of E, the more evenly the microbiota are distributed), H is Shannon-Weiner index, S*_obs_* is the observation species number.

## 3. Results

### 3.1. Microbial Diversity and Community Composition in Sable Gut

We observed that the OTU (Operational Taxonomic Unit) richness index, the Simpson diversity and the evenness of bacteria in wild sable gut was 516.3 ± 299.8 (mean ± SE), 3.16 ± 3.04, and 0.256 ± 0.149, respectively. To determine the relationship between sable gut microbial diversity and the habitat, correlation analysis was performed. Unexpectedly, bacterial community did not respond to the effect of any environmental factor in terms of OTU richness, diversity and evenness.

Multivariate analysis of the bacterial community structures in different groups indicated wild sable gut bacterial community compositions differed between high and low altitude (*p* < 0.05; Figure 2A). Except for altitude, none of environmental factors of fallen trees, hiding cover, canopy density, vegetation coverage, and snow depth were associated with the bacterial community composition.

Sable gut bacterial OTUs were classified into 27 phyla (Appendix A). Firmicutes was the most predominant phylum (accounting for 38.23% of the total sequences), followed by Proteobacteria (30.29%) and Actinobacteria (28.15%) in the whole data. Surprisingly, the above three phyla dominated the vast majority of the sable intestinal microbial community (96.67%), while Bacteroidetes only accounted for 0.47%. Our results also suggested that sable gut microbial community composition was shaped by environmental conditions of host habitat, such as altitude, the number of fallen trees and vegetation coverage (Table 2). We found that altitude significantly influenced the relative abundance of Firmicutes (r = −0.67, *p* < 0.05) and Proteobacteria (r = 0.77, *p* < 0.01). When the effect of surrounding vegetation on bacterial composition was also considered, the proportion of Verrucomicrobia and Gemmatimonadetes were negatively correlated with vegetation coverage (*p* < 0.05), and Spirochaetes and Deferribacteres were enriched in areas where there were more fallen trees (*p* < 0.05).

At genus level, we observed 361 bacterial genera in the sable gut, and the top 20 genera were visualized in graph (Appendix A). Lactobacillus (29.89%) and Pseudomonas (22.39%), the two most frequent genera, represented over half of sequences in sable gut across our dataset. Besides, Arthrobacter, Shigella and Leucobacter were also major sable gut bacteria, accounting for 2.19%, 0.76% and 0.73% sequence frequency. To demonstrate the relationship between environmental conditions and specific genera, correlation analyses were provided (Figure 3A, Appendix A). Similar to phylum level, relative abundance of genera in bacterial communities were change with environmental factors fluctuate indicating that sable gut microbial communities were affected by environmental conditions of host habitat. We observed that relative abundance of Lactobacillus was significantly affected by altitude (r = −0.68, *p* < 0.05), while Pseudomonas (r = 0.67, *p* < 0.05) and Planomicrobium (r = 0.66, *p* < 0.05) showed the opposite trend, with positive correlation with altitude. It is worth mentioning that about 75% of the genera involved in the correlation analyses such as Rhodobacter, Flavobacterium, Allobaculum, Treponema, and Sediminibacterium, were positively correlated with the vegetation coverage of sampling sites (*p* < 0.05), and the canopy density showed a similar effect on some genera such as Humicoccus, Sediminibacterium, Parachlamydia, Pasteuria, Rhodobacter, and Bdellovibrio (*p* < 0.05). Due to the elevation influence was particularly crucial, t-test between high and low altitudes were also performed (Figure 4), which can provide further proof for the above findings. Moreover, we found that Delftia decreased in locations where snow was deeper (*p* < 0.05) and Streptococcus and Akkermansia showed increased abundance in locations with more fallen trees (*p* < 0.05). We also considered the hiding cover effect on bacterial genera, and the result showed that there was a significant correlation between Archromobacter and hiding cover of sampling sites (*p* < 0.01).

Among the above-mentioned genera, we selected four genera, the abundance of which showed correlation with altitude, for linear regression analyses (Figure 5A). As the result showed, the coefficients of determination (r^2^) are 0.517, 0.623, 0.232, and 0.116 of altitude and the relative abundance of Lactobacillus, Pseudomonas, Kurthia, and Olsenella, respectively.

### 3.2. Function Profiles of Sable Intestinal Microbiome

As for the community diversity, also wild sable gut metabolic function was altered in correlation with the altitudes of host habitats (Figure 2B). We observed that altitude significantly (*p* < 0.05) affected functional gene clusters in wild sable gut, while none of the other environmental factors, fallen trees, hiding cover, canopy density, vegetation coverage, or snow depth were associated with the functional gene clusters.

Differenced in altitude was often related to temperature, climate and vegetation type. Changes in altitude, causing variations in living conditions, may bring about adaptations in functions related to metabolism [18,54]. Therefore, correlation analyses between gut microbiota functional genes and environmental factors were performed (Figure 3B, Appendix A). Our results suggested that 20.4% of the total functional genes in wild sable gut responded to altitude changes, and most of them were metabolism-related genes. Among them, the function of Energy metabolism, Lipid metabolism, and Arginine and proline metabolism were stronger at higher locations (r > 0.50, *p* < 0.05), whereas carbohydrate metabolism, nucleotide metabolism, amino acid metabolism, fructose and mannose metabolism, and pyruvate metabolism showed the opposite trend (r < −0.50, *p* < 0.05), indicating an adverse effect of higher elevation on some metabolism-related genes.

In addition to altitude, we also observed that the function of lipoic acid metabolism, caffeine metabolism, and amino acid metabolism were positively affected by vegetation coverage (*p* < 0.05), while functional genes related to transcription factors, bacterial toxins, and ribosome biogenesis in eukaryotes showed negative correlation with vegetation coverage. Our findings suggested that both canopy density and number of fallen trees had positive effect on frequency of functional genes related to fatty acid elongation in mitochondria. Besides, none of the above functional genes in wild sable gut significantly differed with hiding cover changes and fallen trees.

To further explore how altitude changes affected the intestinal metabolic function of wild sables, we included main metabolism-related genes with altitude for linear regression (Figure 5B). The coefficients of determination (r^2^) were 0.618, 0.321, 0.170, 0.730, 0.618, 0.178, 0.714, and 0.443 of altitude for the proportion of energy metabolism, carbohydrate metabolism, lipid metabolism, nucleotide metabolism, fructose and mannose metabolism, amino acid metabolism, arginine and proline metabolism, and pyruvate metabolism, respectively.

Part of pivotal genes were selected for qPCR to reconfirm our above findings. The t-test results revealed that lpxB gene (*p* = 0.037), mlaE gene (*p* = 0.037), and uidA gene (*p* = 0.045) responded to the elevation influence, but not the total bacterial 16S rRNA genes (*p* = 0.099). It was worth mentioning that lipid metabolism related genes lpxB (Figure 4N) and malE (Figure 4O) had greater relative abundance in high altitudes, which corresponded to the trend of KEGG pathway modules Lipid metabolism (Figure 4J) and Lipid biosynthesis proteins (Figure 4K). In contrast, carbohydrate related gene uidA (Figure 4P) were higher in abundance in low altitudes, and it was consistent with the situation of Carbohydrate absorption (Figure 4L).

Our functional gene profile was classified as 7 groups at level 1, 42 functional categories at level 2 and 328 functional genes at level 3 according to the KEGG Ontology (Appendix A). Metabolism was the largest group (47.80% of the total benchmark), including carbohydrate metabolism (11.05%), amino acid metabolism (9.91%), energy metabolism (5.41%), nucleotide metabolism (3.66%), metabolism of cofactors and vitamins (3.63%), lipid metabolism (3.40%), and xenobiotics biodegradation and metabolism (3.05%). Genetic information processing was the second most prominent functional group (17.43%), followed by Environmental Information Processing (17.41%).

## 4. Discussion

### 4.1. Environmental Factors Can Influence Sable Gut Microbial Community

Living conditions can influence wild mammal gut microbial communities in several ways, via different ecosystem type [55], vegetation distribution [56,57], temperature change [21], etc. In our study, bacterial communities in wild sable gut responded to the altitude of host habitat, which support our hypothesis that altitude shapes sable gut microbial community composition. This finding agrees with previous studies, which demonstrated gut microbiota changes across an altitudinal gradient in lizard (*Phrynocephalus vlangalii*) [18] and macaques (*Macaca thibetana*) [58]. In principal, these association could be explained by the comprehensive influence of altitude that involved temperature, oxygen content, vegetation, and animal activities [54]. Despite vegetation and animal diversities in this area descend as altitude increases, surprisingly wild sable gut bacterial diversity did not respond to altitude and other environmental factors. When comparing our results to previous studies on other wild mammals, such as pigs (*Sus scrofa domesticus*) [59] and pikas (*Ochotona curzoniae*) [60] on Qinghai-Tibet plateau, there is a discrepancy that gut microbial diversity of those animals significantly decreased with increasing altitude. The terrain vertical variation is much larger in these areas comparing with our study areas. Here we showed that the relatively small variation in altitude did not charge sable gut microbial community diversity, but the differed vegetation and climate conditions shaped the community composition. Our results cast a new light on wild animal gut that bacterial diversity in sable gut seems to be resilient to environmental changes. This is because sable is a mesocarnivore mammal, which may consume diverse food sources in harsh conditions [11].

### 4.2. Gut Bacterial Structure Varies Along Altitude Increase

The bacteria compositional differences across altitudinal gradient were largely attributable to the distribution of major bacterial taxa in wild sable gut, which also responded to altitude increase. Similarly, altitude has been considered to have significant effects on dominant taxa in house mice (*Mus musculus domesticus*) guts, such as strictly anaerobic bacteria, facultative anaerobes, microaerophiles, and aerotolerant bacteria [54]. Moreover, the relative abundances of some major genera, such as *Akkermansia*, show similar trend across an altitudinal gradient in lizard gut [18]. Our findings revealed that *Lactobacillus*, the most abundant genera in wild sable gut, decreased in high-altitude areas. An explanation for this situation is that *Lactobacillus* mainly process sugar type compounds rather than proteins and lipid [61], but there is less vegetarian resources in elevated habitat where wild sables tend to consume less carbohydrates. Mustonen [15] demonstrated that wild sables may suffer symptoms of hypoglycemia in cold and high-altitude regions where vegetative food sources are scarce, which to a certain extent provides support for our conclusions. Contrary to *Lactobacillus*, another dominant genus *Pseudomonas*, whose relative abundance increased with altitude, has been considered a crucial genus in sable gut with the ability to produce lipase [62]. This finding is consistent with a previous study on black-necked cranes gut microbiota [63]. Among genera with the ability for producing lipase, a focus had been on enzyme produced by members of the genus *Pseudomonas* [64], and the most dominant species of *Pseudomonas* in sable gut was *Pseudomonas fragi*, which is a psychrophilic microorganism and able to produce lipase with high catalytic activity at low temperatures [62]. It is reasonable to conclude that the lower temperature at higher altitudes induced sables demand more low-temperature adaptable lipases to decompose and digest fat, which in turn requires more *Pseudomonas fragi* producing cold active lipases. A previous study on deer mice (*Peromyscus maniculatus*) also suggested that the sustainability of prolonged thermogenesis in high altitude area promotes a shift in metabolic fuel use in favor of lipids over carbohydrates [65]. Our results indicated that sable gut communities mainly adjust the composition ratio of *Lactobacillus* and *Pseudomonas* to regulate metabolic fuel use, so that sables can adapt to the high-altitude environment.

### 4.3. Changes of Gut Microbial Function in Cold Area

The influence of high-altitude can impose thermoregulatory challenges on wild animals, and these stressors may affect both survival and reproduction [65,66]. In this study, dimensionality reduction and correlation analysis results revealed several functional genes enhance as the altitude increased, especially functional groups related to metabolism. These findings led us to accept our second hypothesis, which was consistent with consistent with previous studies. For example, the ability for energy metabolism and fermentation of polysaccharides in house mice gut microbiota was enhanced with increased elevation [54], and gut microbiota of rhesus macaques living at higher altitudes had higher expression of genes related to energy metabolism and lipid metabolism [34]. A possible interpretation is that altitude usually affects other environmental factors such as temperature, oxygen levels, ultraviolet radiation, wind, etc., and it also affects biological factors such as vegetation type, canopy density and concealment level, which in turn affected the physiological characteristics of wild animals [64]. In our study area, the temperature, which was the most relevant and crucial variable related to altitude, significantly decreased from bottom to top, and the maximum temperature difference across this area could reach 15 °C. Therefore, we suggest the direct cause of the enhancement of metabolic functional groups to be the decreasing temperature. Due to the requirement of thermogenesis in low temperature, wild sable gut microbial communities tend to enhance function of energy metabolism. Similarly, Kumar et al. provide support for these findings, that gut microbiota are involved in thermogenesis and energy metabolism during low temperature exposure in mice [67]. Our results also revealed that sable gut microbiota, whose hosts were living in lower temperature, had greater ability to metabolize arginine and proline. This finding is in line with a previous study on oriental fruit fly (*Drosophila melanogaster*), which proposed that gut microbiota played a vital role in promoting the host resistance to low-temperature stress by stimulating its arginine and proline metabolism pathway [68]. In addition to metabolism, the proportion of transcription factor gene also increased when the environment got colder. It is because transcription factors play major roles in regulating the responses to temperature changes [69], sable gut microbial communities improve this ability to cope with temperature changes caused by elevation. Consequently, our findings suggest that wild sables were resilient in adapting to the living conditions, and that altered intestinal microbial communities supported the wild sable to boost various abilities to survive in high-altitude and low-temperature environments.

### 4.4. Wild Sable Survives in Harsh Environment

Anthropogenic activities scattered and decreased the range of forest ecosystem on which wild sables depend, and thus sables were forced to migrate to high-altitude habitats and adapt to a harsh environment. However, due to the robustness and resilience of the gut microbial community, wild sables can still survive under such conditions. Similar conclusions were reached by previous studies on gut microbiota of black-necked cranes (*Grus nigricollis*) [64] and Tibet wild asses (*Equus kiang*) [70] on the Qinghai-Tibet plateau. Taken together, this study shows that wild sables are adaptable to changing conditions and resistant to harsh environments, and that this adaptability is mediated by gut bacterial composition adjustment and metabolic functions enhancement. Despite the relative high-altitude mountains with cold conditions, these poor lands can save wild sables.

We included 10 wild sable gut samples, which is less than in studies on other wild fauna [54,55,71]. However, in our study, snow fall covers most of the sable feces in the forested areas, which intensified our field work load. Comparing with previous effort using three wild sable gut samples [40], our study still enlarged the studied material significantly. The vertical distance between the tallest and lowest samples is around 500 m (from 900 to 1400 m). However, the vegetation and temperature differed between the foot and the top of the mountain area, which formed a clear environmental gradient. In addition, we selected one sample in each of the 10 discrete forest patches with at least 500 m distance from each other. These sampling strategies extended the number and range of wild sable gut samples, as well as minimized the risk of sampling the same sable individual.

## 5. Conclusions

Environmental changes of habitat shaped the sable gut microbial community structure and altered the functions of gut microbiota, showing that the wild sable gut microbial community diversity was resilient and responded to environment change. Elevated habitat is a pivotal factor for wild sable survival and reproduction, and the adaptability is in part enabled through their gut microbial communities. Our observations show that despite having been forced to migrate from low altitudes to high altitudes because of anthropogenic habitat encroachment, wild sables showed toughness in adapting to harsh conditions. Additionally, we propose that the crucial factor enabling wild sables to survive in changeable environments was their gut microbial communities. For example, the adjustment of relative abundance of *Lactobacillus* and *Pseudomonas* regulate metabolic fuel use for wild sables when the food sources differed. It is widely understood that harsh conditions, such as high altitude and low temperature environments, have an adverse effect on wild fauna survival. However, our results suggested that increasing altitude can enhance some functions in wild sable gut microbial communities, such as the abundance of genes related to energy metabolism, lipid metabolism and transcription factors. In the process of land development, space must be reserved as a habitat for wild sables, such as high-altitude mountains and severe cold areas that are not suitable to cultivate.

## Figures and Tables

**Figure 1 animals-11-00865-f001:**
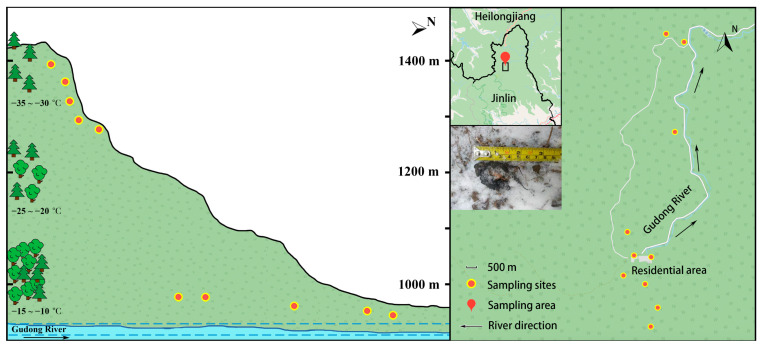
Distribution map and vertical section of sampling sites in Xuexiang National Park in north Jilin, China. Temperature indicates the field temperature range at 1000 m, 1200 m, and 1400 m at night, respectively. Red-Yellow symbol in the right panel indicate approximate altitude of each sampling site within 500 m range.

**Figure 2 animals-11-00865-f002:**
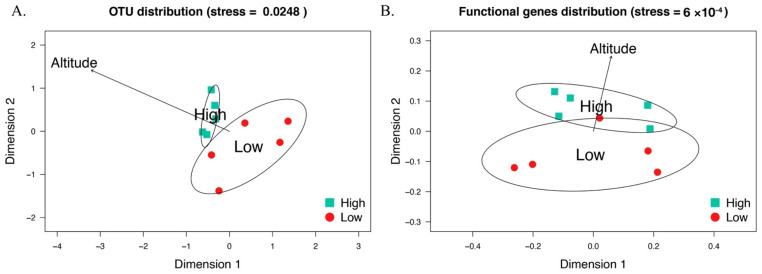
(**A**) Non-metric multidimensional scale (NMDS) plots (based on Bray–Curtis dissimilarity matrix) for bacterial communities. (**B**) NMDS plots (based on Bray–Curtis dissimilarity matrix) for metagenomic functional genes. Statistically significant (*p* < 0.05) environmental variable are shown as arrows using permutation tests by envfit function in vegan package R. The color changes of points from red to blue represents the average altitude of forest patches increases from 900 m to 1400 m.

**Figure 3 animals-11-00865-f003:**
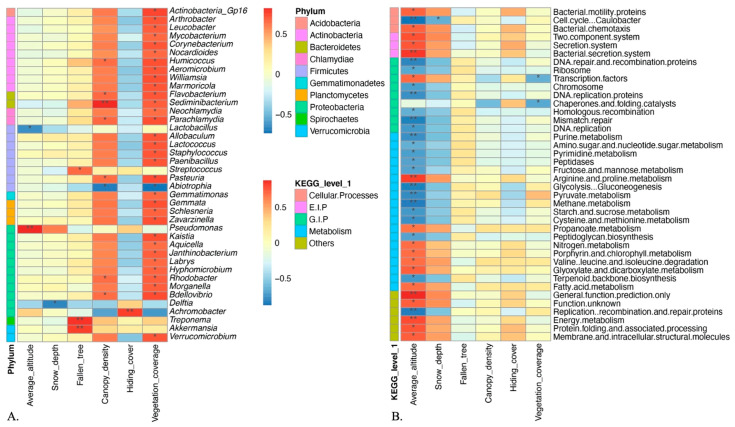
(**A**) Correlation analysis heatmap of bacterial genera and environmental factors. (**B**) Correlation analysis heatmap of pathways and environmental factors. Color changes from blue to red represents the correlation coefficient changes from −1 to 1. Statistically significant (*p* < 0.05 or *p* < 0.01) correlation between phyla and environmental factors are shown as single (*) or double asterisks (**), respectively. E.I.P = Environmental Information Processing, G.I.P = Genetic Information Processing.

**Figure 4 animals-11-00865-f004:**
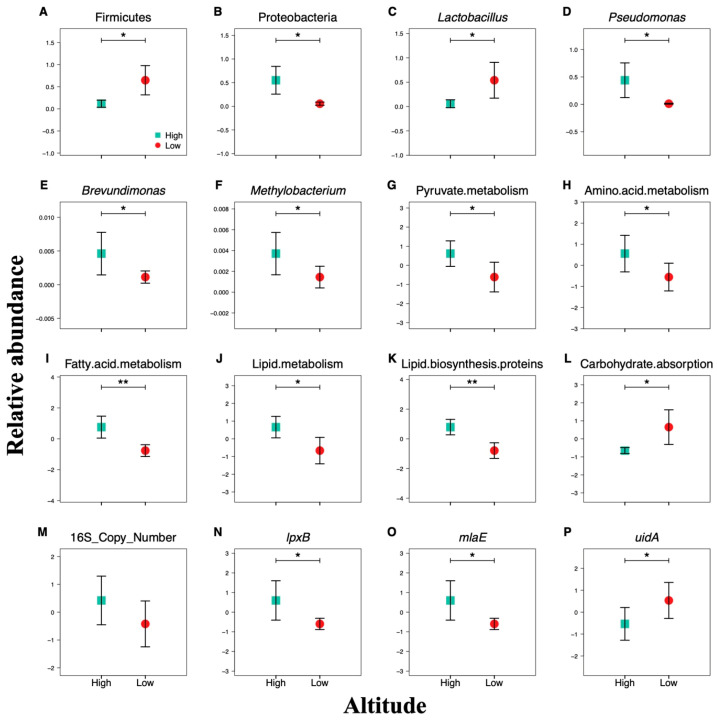
Altitude differences in the relative abundance of major sable gut bacterial phyla ((**A**) Firmicutes; (**B**) Proteobacteria), genera ((**C**) Lactobacillus; (**D**) Pseudomonas; (**E**) Brevundimonas; (**F**) Methylobacterium), KEGG pathway modules at level 3 ((**G**) Pyruvate metabolism; (**H**) Amino acid metabolism; (**I**) Fatty acid metabolism; (**J**) Lipid metabolism; (**K**) Lipid biosynthesis proteins; (**L**) Carbohydrate absorption) and gene copies (qPCR) ((**M**) The total bacterial 16S rRNA genes; (**N**) Lipid A disaccharide synthase related gene, lpxB; (**O**) Lipid asymmetry maintenance ABC transporter permease subunit related gene, mlaE; (**P**) Beta-glucuronidase related gene, uidA). High indicates high altitude group, while Low indicates low altitude group. Asterisks indicate *p*-value of < 0.01 (**) and < 0.05 (*) using t-test.

**Figure 5 animals-11-00865-f005:**
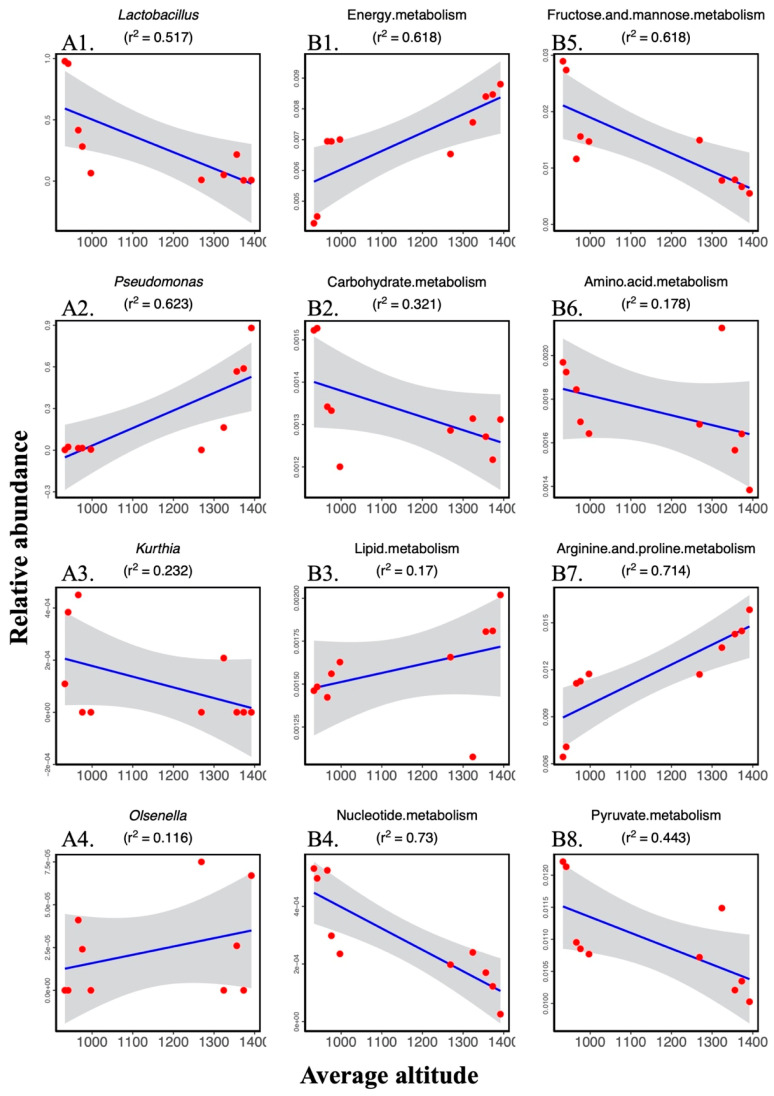
(**A**) Linear regression of the relative abundance of bacterial genera and altitude ((A1) Lactobacillus; (A2) Pseudomonas; (A3) Kurthia; (A4) Olsenella). (**B**) Linear regression of proportion of KEGG pathway modules at level 3 and altitude ((B1) Energy metabolism; (B2) Carbohydrate metabolism; (B3) Lipid metabolism; (B4) Nucleotide metabolism; (B5) Fructose and mannose metabolism; (B6) Amino acid metabolism; (B7) Arginine and proline metabolism; (B8) Pyruvate metabolism).

**Table 1 animals-11-00865-t001:** Geographical information of sampling sites.

Samples	Altitude	Snow Depth (cm)	Vegetation Types	Fallen Tree	Canopy Density	Hiding Cover	Vegetation Coverage
Sable01	Low ^a^	15	Mixed ^c^	0	0.2	0.7	0.3
Sable02	High ^b^	20	Mixed	2	0.6	0.4	0.8
Sable03	Low	20	Broad ^d^	2	0.5	0.4	0.6
Sable05	Low	30	Mixed	1	0.3	0.7	0.5
Sable07	High	35	Mixed	0	0.4	0.7	0.5
Sable08	High	25	Broad	2	0.4	0.8	0.5
Sable11	High	25	Mixed	0	0.3	0.8	0.5
Sable12	Low	20	Broad	1	0.2	0.3	0.3
Sable13	Low	20	Broad	1	0.3	0.7	0.4
Sable16	High	30	Broad	5	0.5	0.7	0.6

^a^ Low: sampling sites with altitude of about 950 m; ^b^ High: sampling sites with altitude of about 1350 m; ^c^ Mixed: Mixed coniferous and broad-leaved forest; ^d^ Broad: Broad-leaved forest.

**Table 2 animals-11-00865-t002:** Correlation analyses between phyla relative abundance and environmental factors.

Phyla	Environmental Factors	*p*	*r*
Firmicutes	Average altitude	0.033	−0.67
Proteobacteria	Average altitude	0.009	0.77
Spirochaetes	Fallen wood	0.004	0.82
Deferribacteres	Fallen wood	0.042	0.65
Verrucomicrobia	Vegetation Coverage	0.036	0.66
Gemmatimonadetes	Vegetation Coverage	0.040	0.65

## Data Availability

The data presented in this study are available in manuscript and Appendix A.

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
