# Peer review of "Habitat Elevation Shapes Microbial Community Composition and Alter the Metabolic Functions in Wild Sable (Martes zibellina) Guts"

_animals, 2021, doi:10.3390/ani11030865_

Round 1

Reviewer 1 Report

A useful and interesting paper that shows the importance of knowledge of the gut microflora to how a species evolves and adapts to habitat change. I think this is a worthy paper for publication.

A few points for consideration.

In the simple summary, the abstract and at the start of the introduction, it would be helpful to explain what a sable is, as some people might think it is the antelope (for example) whereas your paper is about the species of marten. Perhaps simply put the Order and family that it belongs to. 

In the simple summary you say that sable show a toughness at adapting. Is this the sable themselves or their gut microflora? What do you mean by toughness?

Line 59: Is there a reference or evidence for this in the literature?

Line 61: Threatened is not an IUCN Red List category so it might be a threatened species but you need to state what it is exactly, CR, EN, VU or NT. 

Line 84: Whenever a new species is first introduced in the text (e.g. mouse) please don't forget to include the binomial name, i.e. laboratory mouse (Mus musculus domesticus). Please check through the whole paper for this. 

Line 85: Wild bird is very broad. Can you narrow down? 

Line 104: Please explain what you mean by functional enhancement.

Line 106: Assist the host animal?

Line 111: Why typical? Is frigid the right word?

Line 120-125: Can you include common names where relevant. It makes your writing more accessible. 

Line 250: Please explain what data went into the t-test. I.e. a comparison of means (two samples) or the values from one group compared to a published mean (one sample) or a paired t test of repeated measures from the same population?

Line 252: Do your data meet the assumptions for linear regression? Did you do testing for normality, equal variance and how did you determine best model fit?

Line 135: The new habitat being these forested slopes? Please be specific. 

Figure 1: Really clear and useful. Does it need a citation for the map artwork?

Line 234: Can you please define and then explain the formulas for the diversity indices that are stated. Likewise, what does OTU stand for? 

Line 243: Is this 99,999 or 99 & 999 permutations?

Figure 3 is really useful but can you do a simple explanation and pick out some of the most important gut microflora species that correlate with habitat? I.e. if a sable lives at high altitudes 9 out of 10 animals have this bacteria. If a sable lives at low altitudes, 9 out of 10 animals have this bacteria. Sable that move between have... It would help the reader make sense. 

There seems to be a great deal of results presented whereby the analyses is not all explained in the methods. Please revisit the data analysis section of the methods to fully expand on all inferential analyses and how it was applied. For all models, explain what the output variable was and what the potential predictors were. Please lay the data analysis section out in the same order as you plan to set out each result. 

Line 508: think about the word toughness again. What do you mean? Resilience? Adaptability? 

Author Response

Thank you so much for your affirmation and approval for the article. Please see the attachment.

Reviewer 2 Report

The manuscript topic is interesting and it provides  better knowledge of animal microbiome composition and factors affecting this. The sample collection and lab experiments are performed correctly, and material and methods are described adequately. The manuscript is well written and results are presented clearly. Discussion of obtained results is comprehensive.

Remarks

Please avoid using terms 'metagenome' and 'microbiome' as synonyms (P3 L99). P3 L101 replace 'environmental' with 'environment'.

Data analysis - in the case of proportions (OTU, genera, phyla) it is advisable to use clr (centered log-ratio) transformation prior correlation analysis and t-test (this is compositional data).

There is a contradiction in data analysis results. When OTU and phyla data are used then a strong correlation is found between bacterial community and altitude. However, when correlations between top 20 genera and altitude are presented, then only few (5) correlations occur with altitude. Firmicutes and Proteobacteria made up ca 70% of community, and both these phyla were correlated strongly to altitude. Thus, many top 20 genera should belong to these phyla and correlation with altitude could be expected. In the same time, most of the functional properties have strong correlation with altitude. Please check this problem.

Fig 3 legend- (B) shown are not functional genes but pathways on the plot.

Author Response

(The authors gave the same response as above.)

Round 2

Reviewer 1 Report

The authors have done a good job of clarifying key parts of the manuscript, adding in more relevant details and explaining the methods more clearly. I am happy for this manuscript to be published. There is very detailed feedback provided to all of my questions from the first review.

Reviewer 2 Report

I am satisfied with revised manuscript